# The Impact of the Fear of COVID-19 on Purchase Behavior of Dietary Supplements: Integration of the Theory of Planned Behavior and the Protection Motivation Theory

Cheng Liu [1,†], Cheuk-Kwan Sun [2,3,†], Yu-Chia Chang [4,5], Shang-Yu Yang [1], Tao Liu [6] and Cheng-Chia Yang [1,*]

1   Department of Healthcare Administration, College of Medical and Health Science, Asia University, Taichung 41354, Taiwan; 108211012@live.asia.edu.tw (C.L.); henry879019@asia.edu.tw (S.-Y.Y.)
2   Department of Emergency Medicine, E-Da Hospital, Kaohsiung 82445, Taiwan; ed105983@edah.org.tw
3   School of Medicine for International Students, College of Medicine, I-Shou University, Kaohsiung 82445, Taiwan
4   Department of Long Term Care, College of Health and Nursing, National Quemoy University, Kinmen County 892009, Taiwan; ycchang@email.nqu.edu.tw
5   Department of Medical Research, China Medical University Hospital, China Medical University, Taichung 404332, Taiwan
6   Sports Institute, Wuhan Huaxia University of Technology, Wuhan 430223, China; hxutedu0215@163.com
*   Correspondence: chengchia@asia.edu.tw; Tel.: +886-4-23323456 (ext. 20067)
†   These authors contributed equally to this work as first authors.

**Abstract:** This study aimed at assessing the impacts of the fear of COVID-19 on consumer buying behavior toward dietary supplements. This investigation was a cross-sectional study in which literate adults regardless of gender over the age of 20 were recruited from three pharmacies in three different districts of Wuhan City, China. A total of 598 questionnaires were analyzed after excluding 10 with incomplete information. The current study demonstrated that attitudes, subjective norms, and perceived behavioral control had a positive impact on the intention of purchasing dietary supplements. Fear of COVID-19 was related to an enhanced purchase intention toward dietary supplements. Attitudes, subjective norms, and perceived behavioral control were significant factors that mediated the association between the fear of COVID-19 and the purchase intention of dietary supplements. This study helps provide practical advice for stakeholders in the pharmaceutical and healthcare industries to tailor appropriate strategies for improving product promotion or healthcare-related interventions.

**Keywords:** fear of COVID-19; dietary supplements; theory of planned behavior; protection motivation theory

## 1. Introduction

The outbreak of COVID-19 at the end of 2019 triggered unprecedented global socioeconomic unrest [1]. Up to 8 November 2021, there were 250,749,187 people with a confirmed diagnosis of the disease that had claimed up to 5,067,584 lives worldwide [2]. Although the enforcement of lockdown policies in many countries and cities may constitute an effective precaution against viral spread, it has had adverse impacts on people's daily lives as well as their physical and mental health [1].

Compared with other illnesses, one of the important features of infectious diseases (e.g., COVID-19) is the elicitation of fear [3]. During the COVID-19 pandemic, fear is commonly attributed to psychological and emotional stress [4]. For instance, not only are the rapid spread and the uncertainty of transmission media important contributors to people's anxiety but the high infection and mortality rates also create an atmosphere of general fear [5]. Indeed, the fear of contracting the disease through contact with infected individuals has become a chronic, unrelenting psychological burden [6]. From a public

health perspective, the World Health Organization (WHO) recommends precautionary measures against viral dissemination such as washing hands, wearing masks, and maintaining social distance [7]. Besides, the governments of a number of countries have imposed lockdowns and curfews as well as isolation and home quarantine to control the spread of the pandemic [8,9].

In addition to the above-mentioned precautions against COVID-19 from a public health perspective [10], recent studies have revealed the beneficial effects of a number of micronutrients on enhancing immunity that helps alleviate the spread of the disease and the fear of contracting it [11]. For instance, the prophylactic and therapeutic potentials of certain micronutrients (e.g., zinc, vitamins C, D, and E) toward COVID-19 have been reported [12,13]. Previous investigations have shown that high-dose vitamin C could suppress respiratory symptoms caused by viral infections [14]. Its high tolerance and lack of notable side-effects have made it the first choice as a nutritional supplement during the pandemic [15]. Moreover, adequate intake of vitamin D could reduce the risk of contracting influenza and COVID-19 as well as the associated mortality rates through its conversion into 25-hydroxycholecalciferol [25(OH)D3] [16]. The findings have underscored the increasing interest among the general public in the prophylactic potentials of dietary supplements against this unprecedented pandemic [12,17]. However, a ban on public assemblies and cessation of industrial operations not only could cripple the global economy but may also lead to a blockade of the food supply chain during the lockdown period, thereby increasing the risk of malnutrition [18].

The theory of planned behavior (TPB), which is one of the most important models in the prediction of individual behaviors, has served as a tool for investigating the attitudes and intentions toward consuming functional foods [19,20]. Other authors have previously identified a number of factors that affect the intention of purchasing dietary supplements (i.e., a category of functional or healthy foods) including brand, price, quality, and self-perception of one's health status [21–25]. On the other hand, whether the emotional turmoil triggered by the COVID-19 pandemic [26] has a role to play in affecting people's attitudes and intentions toward purchasing dietary supplements remains unclear.

In addition, the current study extended the theory of planned behavior model by incorporating the protection motivation theory (PMT) that highlights the triggering of an individual's self-protective behaviors through fear [27]—such as the fear of contracting COVID-19. A previous investigation has shown that fear is a determinant of consumer buying behavior during the pandemic [28]. The finding is consistent with that of another study that demonstrated that fear is an important factor affecting an individual's behavior and attitudes [29–31]. Moreover, people tend to comply with authoritative expectations and restrictions under high-risk circumstances [32]. The fear of COVID-19 has also been reported to increase self-efficacy and perceived behavioral control [32,33]. Emotional response is widely considered to be a contributor to personal protective behavior [34]. Previous studies have shown that fear could promote threat-alleviating behaviors [12,13,35]. For instance, in accordance with the protection motivation theory (PMT), fear arising from an environmental threat (e.g., waste accumulation) would enhance an individual's participation in the corresponding management behaviors [34]. Through the incorporation of PMT and the extended theory of planned behavior (TPB), a recent study demonstrated a positive impact of perceived vulnerability and perceived severity on the intention to follow rules [36]. Recent investigations utilized the PMT framework to explain consumers' behavioral intentions during the COVID-9 pandemic [37,38]. One of the studies showed that perceived threat and response efficacy could contribute to fear, which is a predictor of consumer behavior during the COVID-19 pandemic [37]. Furthermore, self-efficacy has been found to indirectly reinforce the positive effect of perceived severity of intention to make unusual purchases [38].

Therefore, the current study aimed at assessing the impacts of the fear of COVID-19 on consumer buying behavior toward dietary supplements, focusing on (a) the influences of attitudes, subjective norms, and perceived behavioral control; (b) the question of

whether the fear of COVID-19 would affect the intention to purchase dietary supplements; and (c) the effect of the fear of COVID-19 on attitudes, subjective norms, and perceived behavioral control.

The present study attempted to address three theoretical and practical aspects of the issue. First, we developed a general model to investigate the ways in which COVID-19-associated fear, as well as changes in attitudes, subjective norms, and perceived behavioral control, affected the intention behind consumer buying behavior toward dietary supplements. Second, we tried to bridge the gap of knowledge between the fear of COVID-19 and the intention of purchasing dietary supplements. Finally, through an investigation into the impact of the fear of COVID-19 on the consumer buying behavior toward dietary supplements in the era of the pandemic, the current study helps provide practical advice for stakeholders in pharmaceutical and healthcare industries to tailor appropriate strategies for improving product promotion or healthcare-related interventions.

## 2. Background

### 2.1. Theory of Planed Behavior and Purchase Intention

TPB, which was first proposed by Ajzen in 1985 [39], comprises attitudes (AT), subjective norms (SN), perceived behavioral control (PBC), and behavioral intention (BI). Behavioral intention, which refers to the degree of inclination of an individual to participate in specific activities, can be assessed by the amount of effort that one is willing to put into achieving a goal.

The dependent variable of the current study (i.e., behavioral intention) was purchase intention, which can be defined as the probability that a consumer is willing to take a specific purchase action [40]. Purchase intention, which reflects the motivational intention of purchasing a product, is a validated predictor of buying behaviors [39,41,42]. Previous studies have demonstrated that attitudes, subjective norms, and perceived behavioral control directly or indirectly affect purchase intention [36,43–46].

Attitude is an individual's positive or negative evaluation of a specific behavior. The more positive the attitude, the higher the behavior intention, and vice versa. A number of previous studies have shown a positive association between attitudes and purchase intention [47,48]. For instance, a previous study on foods enriched with omega-3 fatty acids demonstrated that attitudes had the greatest influence on intentions but not subjective norms or control beliefs [19].

Subjective norms, which refer to an individual's perception of approval and support from an important person (e.g., spouse) or a group of people (e.g., parents, friends, and colleagues) regarding a particular behavior, are determined by the perceived social pressure to comply with those people's views. The more positive the subjective norms are, the stronger the behavioral intention (e.g., purchase intention) becomes. Previous investigations have highlighted the direct impact of subjective norms on the purchase behavior of dietary supplements [49,50].

Perceived behavioral control is defined as an individual's perception of the opportunity and difficulty in enacting a behavior after taking into consideration the relevant factors (e.g., available resources and skills). Therefore, those who have better abilities, knowledge, and supplies of resources are likely to have a stronger perceived behavioral control than that those who consider the behavior difficult [39]. A previous study has identified perceived behavioral control as an essential contributor to purchase intention [50]. Another study on Chinese customers who purchased imported soy-based dietary supplements demonstrated consistent findings by showing an active role of perceived behavioral control in enhancing purchase attitudes [51]. Therefore, the current study proposed the following hypothesis:

**Hypothesis (H1).** *Intention antecedents are positively associated with purchase intention, including the attitudes toward dietary supplements (H1a), subjective norms (H1b), and perceived behavioral control (H1c).*

### 2.2. The Fear of COVID-19 and Purchase Intention

Emotions (e.g., fear) have been found to play a critical role in consumer behaviors when faced with the threat of COVID-19 [37]. Fear of COVID-19 refers to the anxiety, depression, and other negative emotional impacts triggered by COVID-19 [52]. Fear, which is defined as an unpleasant mental status elicited by a threat or stimulus [53], is one of the primitive human emotions associated with an instinctive response critical for survival [54]. Appealing to such a primitive emotion through harnessing fear for propaganda and advertisement has been found to be highly effective for boosting customer purchase intention [55].

Fear, in the forms of perceived vulnerability to threats and perceived severity of threats, has been shown to augment behavioral intention. During the COVID-19 pandemic, fear generated from individuals' cognitive evaluations of the threat and their ability to engage in risk preventative actions were significant indicators of the observed customer behaviors related to their restaurant visits [37]. A previous study also demonstrated that consumers are more likely to be attracted by products that would reduce the risk of being infected when shrouded in fear of the pandemic [56]. Besides, an appeal to fear could be positively associated with purchase behavior toward selected personal protective equipment [31,57]. On the other hand, fear has also been shown to have an adverse impact on purchase intentions. For instance, a previous study has revealed a significant negative correlation between the fear of crime at the shopping site and purchase intentions [58]. Moreover, fear of the fraudulent use of electronic information (e.g., identity, credit card number) while shopping on the internet has also been reported to negatively affect consumers' buying intentions [59]. Taken together, fear could be an important factor for extending the TPB model to better understand the purchase intentions of consumers during the pandemic. Hence, the present study hypothesized that:

**Hypothesis (H2).** *Fear of COVID-19 is positively associated with one's purchase intention toward dietary supplements.*

### 2.3. Integration of Fear of COVID-19 and the Theory of Planned Behavior

A recent investigation has revealed a positive association between the degree of fear of COVID-19 and the attitudes to and intention of purchasing facial masks [57]. Based on such findings, the current study hypothesized that the degree of fear of COVID-19-induced adverse events among consumers would be positively correlated with their attitudes towards purchasing dietary supplements that could prevent them from contracting the disease. The change in attitudes would, in turn, positively influence their purchase intentions [13,60].

A previous study has highlighted that subjective norms have a crucial role to play in determining whether the public would comply with the policy of mask-wearing [61]. Under critical circumstances (e.g., facing the risk of COVID infection), consumers are likely to follow the advice given by those important to them [62]. In other words, people tend to comply with the expectations or restrictions imposed on them by authoritative figures [32]. Accordingly, the current investigation hypothesized a positive association between the degree of consumers' fear of the pandemic and their compliance with the advice given by those deemed important to them. The recommendation made by those important figures could have a significant influence on consumers' purchase intention toward a specific product [63].

In addition, the public fear of COVID-19 could raise concerns about their own immunity, and they may thus choose to use dietary supplements (e.g., vitamins) in order to minimize their chances of infection [12,13]. In this way, their perceived behavioral control may be enhanced [32,33]. Therefore, the stronger their fear of COVID-19, the more likely they would collect information about the prophylactic effects of dietary supplements against the disease as well as intend to gain access to related resources. Possessing the relevant information and gaining access to the resources would, in turn, boost their purchase intentions. Based on these arguments, the present study hypothesized that:

**Hypothesis (H3).** *Fear of COVID-19 is positively associated with attitudes toward dietary supplements (H3a), subjective norms (H3b), and perceived behavioral control (H3c).*

Furthermore, this study attempted to elucidate the associations between attitudes, subjective norms, and perceived behavioral control with the fear of COVID-19 and purchase intentions of dietary supplements. According to PMT, individuals' fear would change their behavioral responses to specific events or outcomes to minimize the resulting trauma [64]. PMT, which is a theory on preventive health behavior, has been widely applied to the investigation of intentions and behaviors in a variety of health-related disciplines [64]. PMT assumes that individuals participate in risk-avoiding behaviors based on their incentive of self-protection against the threat of infection [64]. A previous report has shown risk-avoiding behaviors (e.g., wearing masks, keeping social distance, regularly washing hands) in individuals with a fear of COVID-19-related adverse consequences [65].

The current investigation further hypothesized that, based on PMT, people would pay attention to dietary supplements that would reinforce their immunity (e.g., vitamins C, D, E) after realizing the importance of their own immunity due to their fear of COVID-19. Therefore, this study proposed that the components of TPB, including attitudes, subjective norms, and perceived behavioral control, are important mediators in the purchase intention of dietary supplements among those in fear of the pandemic. As mentioned above, this study hypothesized a cascade triggered by the dissemination of COVID-19, followed by fear of the pandemic that results in a realization of the importance of self-protection. As one of the protection-seeking behaviors, consumers would tend to adopt a positive attitude toward dietary supplements with an enhanced purchase intention on the belief that the products would be beneficial for the prevention of COVID-19. They would also acquire related knowledge and attempt to gain access to those products. Moreover, people are likely to seek and be influenced by the opinions of those considered to be important to them during critical decision-making. Therefore, the present study hypothesized that:

**Hypothesis (H4).** *The attitudes toward dietary supplements (H4a), subjective norms (H4b) and perceived behavioral control (H4c) mediate the relationship between fear of COVID-19 and purchase intention.*

## 3. Methods

### 3.1. Definition

For the current study, dietary supplement referred to a product taken by mouth that consists of a "dietary ingredient" intended to supplement the diet, including vitamins, minerals, amino acids, herbs or other botanicals as well as substances such as enzymes, glandulars, metabolites, and organ tissues according to the Dietary Supplement Health and Education Act (DSHEA) of 1994 of the Food and Drug Administration (FDA) of the United States [66]. Dietary supplements also include concentrates or extracts and may take different forms (e.g., tablets, liquids, powders, capsules, gelcaps, softgels). Moreover, one product may contain one or more supplements [67]. A dietary supplement is supposed to serve the purpose of health maintenance and disease prevention [68].

### 3.2. Study Participants, Protocol, and Procedures

Using the convenience sampling approach, the current cross-sectional structured questionnaire-based study collected information from literate adults regardless of gender over the age of 20 in three pharmacies over three different districts (i.e., Qiaokou, Hongshan, and Wuchang) of Wuhan City, China. All study protocols and procedures were reviewed and approved by the institutional review board of Taichung Jen-Ai Hospital (Approval No.: 110-25). Between January 3 and 31, 2021, a total of 608 participants gave their informed consents and completed a questionnaire designed for the current project after listening to the oral explanation of the researchers. The incentive was a 10 Chinese Yuan coupon for each participant after completion of the questionnaire. Before filling in the questionnaire, all

participants were shown pictures of dietary supplements and given full definitions of such products (e.g., minerals, vitamins C, D, E, and probiotics). A total of 598 questionnaires were analyzed after the exclusion of 10 with incomplete information. In addition, some researchers suggest that the number of subjects should be at least ten times that of the questionnaire items (i.e., 20 in this current study) [69]. In this way, the minimal number of respondents was 200 for the present study, which recruited a total of 598 participants. Therefore, the final sample size was deemed acceptable for achieving statistically significant results.

### 3.3. Questionnaire and Study Parameters

The questionnaire consisted of three sections. The first section involved a TPB assessment, while the second and third sections focused on the evaluation of the fear of COVID-19 and demographic data (i.e., age, gender, marital status, education level, monthly salary, and purchase experience), respectively. The whole questionnaire was completed by the participant.

For the current study, the questionnaire was designed to investigate different aspects of TPB according to the framework of a previous study [41], including the attitudes, subjective norms, perceived behavioral control, and behavioral intention. There were three questions for each aspect of this theory. For assessing the "behavioral attitude", i.e., whether the participant had a positive or negative attitude toward dietary supplements, they were presented with the statements: (1) Dietary supplements are helpful for me; (2) The purchase of dietary supplements is mandatory for me; (3) Buying dietary supplements is a very sensible choice for me. Furthermore, there were three statements on "subjective norms" to evaluate the degree of support regarding the use of dietary supplements from someone important to the participant (e.g., family and friends): (1) My family supports my purchase of dietary supplements; (2) My friends support my purchase of dietary supplements; (3) People important to me support my purchase of dietary supplements. For "perceived behavioral control" that focused on evaluating the adequacy of knowledge and resources as well as the degree of self-control regarding using dietary supplements, the three statements were: (1) The decision to purchase dietary supplements is solely on me; (2) I have the resources, time, and opportunity to purchase dietary supplements; (3) I will purchase dietary supplements if I feel I have the need. There were three statements assessing the "behavioral intention" toward the purchase of dietary supplements, including: (1) I am willing to purchase dietary supplements for maintaining a healthy condition; (2) I have suggested that others buy dietary supplements; (3) I am willing to pay more for dietary supplements than for ordinary food products. To address the impact of fear of COVID-19 on the purchase of dietary supplements, we adopted the COVID-19 Phobia Scale (C19P-S) from a previous study [4] that involves the evaluation of effects from four main categories of factors, namely, psychological factors, social factors, psychosomatic factors, and economic factors. Of the four categories, economic factors evaluate the fear arising from a COVID-19-induced shortage of food and resources, while psychosomatic factors aim to assess whether people experience various psychosomatic difficulties such as stomachache. Nevertheless, taking into consideration that the shortage of resources and food were already alleviated at the time of this survey compared to that at the beginning of the pandemic as well as the fact that COVID-19-related symptoms were notably relieved after effective control of disease transmission, the current study deleted the categories of economic and psychosomatic factors from its original design and focused on the evaluation of the impacts of psychological and social factors. There were six statements to assess the fear of COVID-19 at the psychological level, such as "I feel anxious about contracting COVID-19" and "I worry that my family may get COVID-19". On the other hand, there were five questions for evaluating the degree of anxiety arising from social interactions following the outbreak of COVID-19, including "I feel worried when I see people coughing since the COVID-19 outbreak" and "I try my best to avoid seeing people sneezing during the pandemic". All items in the questionnaire were scored on a five-point Likert scale with

"1" representing "strongly disagree" and "5" denoting "strongly agree" (see Supplementary Materials).

To validate the questionnaire, six experts (i.e., four experts from relevant industries and two professors from academic institutes) were invited to review the preliminary version and provide suggestions. Every item in the questionnaire was given a score by the experts regarding its necessity and suitability for being included. Language amendments were also made. The content validity index (CVI) of the final version was found to be 0.964, which was higher than the required value of 0.8 [70].

### 3.4. Statistical Analysis

We used a two-stage procedure to perform SEM analysis with AMOS 19.0. In the first stage, we established the quality and adequacy of measurement through CFA by ensuring reliability and convergent and divergent validity. Then, we used SEM to test the causal relationships between the latent variables in the second stage. In each stage, a maximum likelihood estimation method was employed. Assessment of goodness-of-fit was made by multiple indicators: $\chi2$ (chi-square), $\chi2/df$ (chi-square to degree of freedom ratio), CFI (comparative fit index), GFI (goodness-of-fit index), TLI (Tucker–Lewis index), and RMSEA (root mean square error of approximation).

## 4. Results

### 4.1. Demographic Characteristics of Participants

A total of 598 participants were recruited for the current study. There were slightly more females than males (Female: n = 342, 57.2%; Male: n = 25, 42.8%). The mean age of participants was 43 years with the highest proportion of individuals aged between 40 and 50 (n = 210, 35.1%), followed by those aged between 30 and 40 (n = 153, 25.6%). With respect to marital status, the majority of the participants were married (n = 469, 78.5%). Regarding educational level, 70% of the subjects were university graduates (n = 419), followed by high school or occupational college graduates (n = 114, 19.4%). The average monthly salary of the participants was 5984 Chinese Yuan. Experience of purchasing dietary supplements was noted in 436 (73%) of the recruited individuals (Table 1).

**Table 1.** Personal demographic characteristics.

| Attributes | Distribution | Frequency | % |
|---|---|---|---|
| Gender | Male | 256 | 42.8% |
| | Female | 342 | 57.2% |
| Age | 21–30 years | 91 | 15.22% |
| | 31–40 years | 153 | 25.59% |
| | 41–50 years | 210 | 35.10% |
| | Above 50 years | 144 | 24.08% |
| Education level | Junior high school or below | 7 | 1.17% |
| | Senior high school | 114 | 19.06% |
| | College | 419 | 70.07% |
| | Master's degree or above | 58 | 9.70% |
| Marital status | Married | 278 | 46.49% |
| | Single | 319 | 53.34% |
| | Divorced/widowed | 1 | 0.17% |
| Monthly income (Chinese yuan) | 2000 or below | 33 | 5.52% |
| | 2001–4000 | 189 | 31.61% |
| | 4001–6000 | 207 | 34.62% |
| | 6001–8000 | 103 | 17.22% |
| | 8001 or above | 66 | 11.04% |
| Buying experience (Within a month) | Yes | 436 | 72.9% |
| | No | 162 | 27.1% |

To prevent and mitigate the problem of common method variance (CMV), the present study employed pretest prevention and post-test detection [71]. Pretest prevention was achieved by anonymous completion of the questionnaire by the participants, while six factors with eigenvalues over 1 under unrotated circumstances were extracted using Harman's single factor test for post-test detection. Our computation showed a cumulative explained variance of 59.6% and a first factor explained variation of 39.31% (i.e., less than 50%). Hence, the preliminary determination showed a non-significant CMV effect.

*4.2. Reliability and Validity of Study Instrument*

Fit indices of the initial measurement model showed that the model did not meet the required criteria for model fit indices for CFI. Observed variables with factor loading of less than 0.50 were removed as recommended by a previous study [72]. Hence, two psychological factors (PSF5, PSF6) and one social factor (SF3) were deleted. Following this procedure, all model fit measures were satisfied (CMIN/DF = 3.9, CFI = 0.948, GFI = 0.905, AGFI = 0.876, NFI = 0.931, TLI = 0.938 and IFI = 0.948; RMSEA, 0.07).

The measurement model of this study's reflective indicators was appraised by calculating the individual item reliability, composite reliability (CR), and average variance extracted (AVE). The CR of the latent variables was the composite of the reliability of all the measurement variables and represented the internal consistency of the constructed index. Higher reliability indicated higher internal consistency of the latent variables. A previous study [73] recommended a CR equal to or greater than 0.7; Table 2 demonstrates a value of CR for each variable in the current study between 0.8760 and 0.936 (i.e., greater than the 0.7 standard), denoting favorable internal consistency. The power of each measurement variable was indicated by the AVE of the latent variables; a higher AVE indicated higher discriminatory validity and convergent validity of a latent variable. A published study [74] recommended an AVE greater than 0.5. In the current study, the AVEs of the latent variables were all between 0.62 and 0.819 (Table 2) (i.e., greater than the 0.5 standard value) and values of MSV were all found to be lesser than AVE, suggesting favorable convergent validity of the reflective measurement variables.

**Table 2.** Convergent and reliability validity of reflective metrics.

| Parameter | Mean | SD | Item | Loading | T-Value | CR | AVE | MSV | Cronbach's Alpha |
|---|---|---|---|---|---|---|---|---|---|
| Attitudes (AT) | 3.33 | 0.87 | AT1<br>AT2<br>AT3 | 0.902<br>0.883<br>0.917 | 54.077<br>54.2<br>78.931 | 0.928 | 0.811 | 0.729 | 0.875 |
| Subjective norms (SN) | 3.37 | 0.86 | SN1<br>SN2<br>SN3 | 0.904<br>0.909<br>0.903 | 77.041<br>78.691<br>72.036 | 0.931 | 0.819 | 0.729 | 0.914 |
| Perceived behavioral control (PBC) | 3.51 | 0.82 | PBC1<br>PBC2<br>PBC3 | 0.757<br>0.884<br>0.869 | 17.68<br>69.87<br>62.921 | 0.876 | 0.704 | 0.652 | 0.820 |
| Behavioral intention (BI) | 3.34 | 0.86 | BI1<br>BI2<br>BI3 | 0.878<br>0.88<br>0.875 | 62.674<br>71.352<br>53.411 | 0.909 | 0.77 | 0.595 | 0.864 |
| Fear of COVID-19 | 3.42 | 0.79 | PSF1<br>PSF2<br>PSF3<br>PSF4<br>SF1<br>SF2<br>SF4<br>SF5 | 0.775<br>0.794<br>0.734<br>0.801<br>0.823<br>0.808<br>0.758<br>0.833 | 36.461<br>38.608<br>25.147<br>35.22<br>44.479<br>42.846<br>24.209<br>35.797 | 0.931 | 0.626 | 0.618 | 0.936 |

CR: composite reliability; AVE: average variance extracted; MSV: maximum share variance.

Finally, discrimination validity was assessed by computing the square root of the AVE. In the same construct, a square root greater than the other coefficients indicated a weaker relationship among the latent constructs compared to those within the construct, supporting the favorable discrimination validity of the measurement model. The present study had a computed square root greater than the coefficients of each dimension (Table 3), signifying a high discrimination validity of the dimensions.

**Table 3.** Matrix of latent constructs in the measurement model.

|  | Mean | SD | 1 | 2 | 3 | 4 | 5 |
|---|---|---|---|---|---|---|---|
| AT | 3.33 | 0.87 | **(0.901)** |  |  |  |  |
| SN | 3.37 | 0.86 | 0.854 *** | **(0.905)** |  |  |  |
| PBC | 3.51 | 0.82 | 0.808 *** | 0.741 *** | **(0.839)** |  |  |
| BI | 3.34 | 0.86 | 0.71 *** | 0.686 *** | 0.772 *** | **(0.877)** |  |
| Fear of COVID-19 | 3.42 | 0.79 | 0.774 *** | 0.786 *** | 0.751 *** | 0.703 *** | **(0.791)** |

The bold components are the square roots of AVE values and the others are the correlation coefficients. AT: Attitudes; SN: Subjective Norms; PBC: Perceived behavioral control; BI: Behavioral intention; ***: $p < 0.001$.

A previous study [75] exhibited the use of heterotrait–monotrait (HTMT) for estimating the correlations between different constructs. For HTMT, a correlation coefficient less than 0.9 signifies adequate discriminant validity [76]. Our results demonstrated that the correlation coefficients among the constructs were all less than 0.9, indicating satisfactory discriminant validity (Table 4).

**Table 4.** Heterotrait–monotrait (HTMT) between study constructs.

|  | AT | SN | PBC | BI |
|---|---|---|---|---|
| SN | 0.895 |  |  |  |
| PBC | 0.873 | 0.859 |  |  |
| BI | 0.869 | 0.837 | 0.881 |  |
| Fear of COVID-19 | 0.495 | 0.470 | 0.548 | 0.609 |

AT: Attitudes; SN: Subjective Norms; PBC: Perceived behavioral control; BI: Behavioral intention.

### 4.3. Mediation Regression Models of Study Variables

After confirming the measurement model, a structural model was tested to assess the causal relationships between latent variables [77]. Each path can be considered statistically significant and supported if the path coefficient is greater than 1.96 and the probability value is less than 0.05 [77]. Model fit of the structural model was satisfied (CMIN/DF = 3.18; CFI = 0.894; GFI = 0.885; AGFI = 0.86; CFI = 0.87; RMSEA = 0.07). An assessment of path coefficients (Table 5) revealed that attitudes, subjective norms, perceived behavioral control, and fear of COVID-19 all significantly affected purchase intention, thus confirming H1a, H1b, H1c, and H2, respectively.

**Table 5.** Results of the structural equation model and hypothesis testing.

| Hypothesis | Path | Estimate | S.E. | C.R. | Supported |
|---|---|---|---|---|---|
| H1a | AT → BI | 0.240 *** | 0.033 | 7.171 | Yes |
| H1b | SN → BI | 0.284 *** | 0.033 | 8.581 | Yes |
| H1c | PBC → BI | 0.384 *** | 0.040 | 9.622 | Yes |
| H2 | Fear of COVID-19 → BI | 0.159 ** | 0.055 | 2.873 | Yes |
| H3a | Fear of COVID-19 → AT | 0.699 *** | 0.057 | 12.35 | Yes |
| H3b | Fear of COVID-19 → SN | 0.666 *** | 0.055 | 12.089 | Yes |
| H3c | Fear of COVID-19 → PBC | 0.757 *** | 0.06 | 12.537 | Yes |

AT: Attitudes; SN: Subjective Norms; PBC: Perceived behavioral control; BI: Behavioral intention; S.E.: standard deviation; C.R.: Composite Reliability; **: $p < 0.01$; ***: $p < 0.001$.

The results of the positive and direct effect of attitudes on purchase intention (standardized direct effect β = 0.24, $p < 0.05$) were statistically significant, so H1a was supported. The results of the positive and direct effect of subjective norms on purchase intention (standardized direct effect β = 0.284, $p < 0.05$) were statistically significant, thereby supporting H1b. In addition, the significant impacts of both perceived behavioral control (standardized direct effect β = 0.384, $p < 0.05$) and fear of COVID-19 (standardized direct effect β = 0.159, $p < 0.05$) on purchase intention supported H1c and H2, explaining 80.3% of the variance in behavioral intention ($R^2 = 0.803$).

An assessment of path coefficients (Table 5) revealed that fear of COVID-19 significantly affected attitudes, subjective norms, and perceived behavioral control, thereby confirming H3a, H3b, and H3c, respectively. The results of the positive and direct effect of fear of COVID-19 on attitudes (standardized direct effect β = 0.699, $p < 0.05$) were statistically significant, so H3a was supported. That explained 32.3% of the variance in attitude ($R^2 = 0.323$). The results of the positive and direct effect of fear of COVID-19 on subjective norms (standardized direct effect β = 0.666, $p < 0.05$) were statistically significant; therefore, H3b was supported. That also accounted for 30.3% of the variance in subjective norms ($R^2 = 0.303$). The statistically significant positive and direct effect of fear of COVID-19 on perceived behavioral control (standardized direct effect β = 0.757, $p < 0.05$) was in support of H3c. That explained 37.3% of the variance in perceived behavioral control ($R^2 = 0.373$).

The study used previously described steps [77] to apply Preacher & Hayes' approach to the mediation model. First, the study confirmed the direct effect between fear of COVID-19 and behavioral intention. This effect was positive and significant (β = 0.529, t = 12.106; $p < 0.001$; Figure 1). The second step included the effect of the mediator variable (AT, SN, PBC). The indirect effect was positive and significant (H1, H2, H3 were supported; Figure 1). The mediating effect did suppress the direct effect as reflected by the direct relationship between fear of COVID-19 and behavioral intention (β = 0.159, t = 2.87; $p < 0.05$). The study assessed the indirect effects using the bootstrap procedure previously reported [77]. If the 95% CI of the mediation effect did not contain 0, the mediation effect was significant (i.e., existence of a mediation effect). The effect of fear of COVID-19 on behavioral intention through attitudes was 0.155 (standard error (SE) = 0.064, 95% CI (0.042, 0.262)). The effect of fear of COVID-19 on behavioral intention through subjective norms was 0.175 (SE = 0.058, 95% CI (0.079, 0.301)). The effect of fear of COVID-19 on behavioral intention through perceived behavioral control was 0.270 (SE = 0.066, 95% CI (0.161, 0.407)). The three paths did not contain zero, suggesting that mediation effects existed and H4a, H4b, and H4c were supported (Table 6).

**Table 6.** Direct, indirect and total effects of the SEM components.

| Model Pathways | Effect | 95% Boot *CI* |
|:---:|:---:|:---:|
| Direct Path | | |
| Fear of COVID-19 → BI | 0.159 | (0.031–0.272) |
| Indirect Path | | |
| Total: | 0.601 | (0.481–0.737) |
| Fear of COVID-19 → AT → BI | 0.155 ** | (0.042–0.262) |
| Fear of COVID-19 → SN → BI | 0.175 *** | (0.079–0.301) |
| Fear of COVID-19 → PBC → BI | 0.270 *** | (0.161–0.407) |

AT: Attitudes; SN: Subjective Norms; PBC: Perceived behavioral control; BI: Behavioral intention; **: $p < 0.01$; ***: $p < 0.001$.

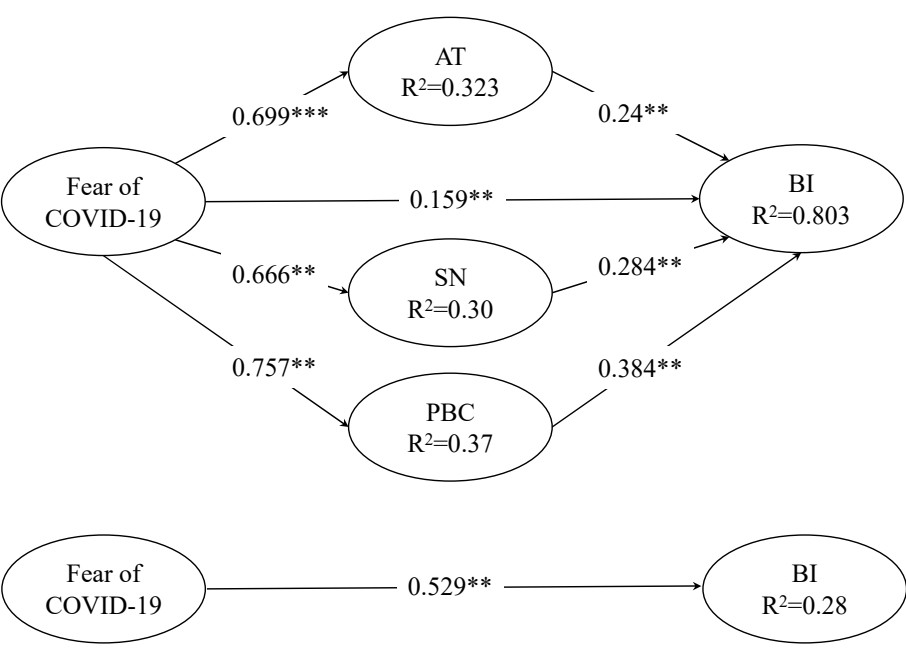

**Figure 1.** Research Structure Diagram. AT: Attitudes; SN: Subjective Norms; PBC: Perceived behavioral control; BI: Behavioral intention; **: $p < 0.01$; ***: $p < 0.001$.

## 5. Discussion

Although disruption of the global supply chain by the COVID-19 pandemic has inflicted substantial negative socio-economic turmoil worldwide [18], its influence on consumers' purchase behavior of dietary supplements has not been adequately addressed. Gaining an in-depth understanding of changes in consumers' behavior through a theoretical framework is crucial to the marketing promotion of relevant products. Through combining the protection motivation theory (PMT) with the theory of planned behavior (TPB), this study investigated the impact of the COVID-19 pandemic on the intention to purchase dietary supplements among the general public under the threat of COVID-19. To the best of our knowledge, the present study is the first to address this issue. Our results, which shed light on the effect of the fear of COVID-19 on the psychological decision-making process defined by the TPB, are expected to be practical for promoting consumers' purchase intention of dietary supplements.

### 5.1. Theory of Planned Behavior (TPB) and Purchase Intention

Our findings showed that the inclusion of the fear of COVID-19 in TPB could at least partly explain the consumers' purchase intention of dietary supplements. The results also supported the proposed model of the current study and the four hypotheses by demonstrating a positive association of purchase intention toward dietary supplements with the participants' attitudes, subjective norms, and perceived behavioral control. In other words, the purchase intention of dietary supplements would increase with a positive attitude, a strong subjective norm, and a high perceived behavioral control.

First, our finding of a positive association between consumers' attitude (AT) and purchase intention was consistent with that of published studies. A previous investigation showed a positive correlation between purchase attention and the attitudes toward a product [78]. Therefore, active promotion of consumers' recognition of a product such as emphasizing the advantages of a dietary supplement and enhancing their knowledge of and confidence in that product would positively impact their attitudes [79].

Second, the results of the present study showed a positive correlation between consumers' subjective norms (SN) and their purchase intention. Consistently, a similar positive association has been reported between the purchase intention of vitamin supplements and subjective norms in a non-COVID-19 setting [80]. In addition, a previous investigation has

demonstrated a positive influence of consumers' prior knowledge of utilizing renewable energy by peers on their purchase behavior [46].

Finally, regarding the finding of a positive correlation between perceived behavioral control (PBC) and purchase intention, a previous study has highlighted a beneficial impact of perceived behavioral control on the intention of purchasing wellbeing food [81]. Consistently, an investigation has demonstrated that the purchase intention toward a health product is usually increased with an enhanced recognition of the importance of physical wellbeing [82].

### 5.2. Fear of COVID-19 and Purchase Intention

The present study showed a positive impact of the fear of COVID-19 on consumers' purchase intention toward dietary supplements. Recent studies have shown that the emotional response (e.g., fear) triggered by the perceived threat from COVID-19 could influence consumers' behavior [36,37]. In particular, fear has been found to affect the perception, choice, and purchase of a product [37,38]. For instance, the fear and uncertainty of COVID-19 have enhanced consumers' environmental concerns and their green hotel brand trust, thereby increasing their willingness to pay more and make sacrifices to stay at green hotels [83]. Moreover, previous studies have demonstrated a tendency toward purchasing green products to reduce carbon emissions when people are exposed to the fear of climate change and air pollution [12,13].

Wuhan, where the first outbreak of COVID-19 occurred, was the first city to implement a lockdown measure in an attempt to impede viral dissemination. The globally unprecedented measure affected over 11 million of its residents from all walks of life who were subjected to not only the fear of the disease but also its adverse impact on their daily lives. For instance, a change in dietary habits could cause malnutrition, resulting in a suppressed immunity and an increased susceptibility to viral infection [84]. Following the first outbreak, inhabitants of the city may tend to choose appropriate dietary supplements to reinforce their immunity against infection from another outbreak [85]. This may partly explain the positive association between their degree of fear of the pandemic and their intention to purchase dietary supplements.

### 5.3. Incorporation of the Protection Motivation Theory (PMT) into the Theory of Planned Behavior (TPB)

To address the effect of fear (e.g., that of being infected) on behavior, the current study incorporated the protection motivation theory (PMT) with the theory of planned behavior (TPB). The results showed that the fear of COVID-19 had a positive impact on the attitudes, subjective norms, and perceived behavioral control toward the purchase of dietary supplements. Moreover, the association between the fear of COVID-19 and purchase intention was mediated by the effects of attitudes, subjective norms, and perceived behavioral control. In other words, the realization of a protective role of dietary supplements against COVID-19 among consumers facing the threat of the pandemic [11] had a positive influence on their attitudes and purchase intention toward dietary supplements. The present study also demonstrated a mediating effect of subjective norms on the correlation between the fear of COVID-19 and the purchase intention of dietary supplements. A previous study has shown significant social and subjective impacts on people's decisions [86]. The stronger the subjective norms, the higher the purchase intention [41]. Therefore, on encountering the threat of COVID-19, consumers tend to seek advice from important individuals whose positive recommendation on dietary supplements would enhance the consumers' purchase intention [87].

Furthermore, perceived behavioral control is the most important mediator of the association between the fear of COVID-19 and purchase intention. The fear of COVID-19 triggers the behavior of disease prevention such as concern about one's immunity out of fear of being infected. Therefore, a fear of the pandemic could enhance an individual's behavioral control [33]. Previous investigations have shown a positive association between the fear of the pandemic among consumers and their knowledge and resources pertinent

to dietary supplements which, in turn, would increase their purchase intention [12,13]. Therefore, a fear of the pandemic could enhance an individual's behavioral control [33].

*5.4. Practical Implications*

In the era of the COVID-19 pandemic, the findings of the current study provide important insights for suppliers and marketers of dietary supplements as well as for clinicians. Knowing the impact of the fear of COVID-19 on the purchase intention toward dietary supplements, marketers may implement appropriate strategies for promoting their products. For instance, an emphasis on the potential immune-reinforcing benefits of dietary supplements may appeal to consumers in need of treatment or prevention of COVID-19. Previous investigations have proposed the dissemination of information on the efficacies of certain dietary supplements (e.g., vitamins C and D) through advertisements, health education, or other marketing approaches to reinforce consumers' trust and confidence in the benefits of those products [12,13]. Such measures could drive consumers toward better health-promoting decisions [88]. In addition, a previous study has shown that an emphasis on the high susceptibility to a disease and its severity is an important determinant of the success of the fear-based product-promoting strategy [89]. Previous research has demonstrated a significant positive impact of arousing fear of a physical condition associated with a product on promoting behavioral intention, such as showing warnings or illustrations on the package of cigarettes (e.g., aging skin and discolored lungs from smoking cigarettes) [54]. Despite the lack of pharmacological action, dietary supplements still play an important role in the regulation of physiological functions. Through different means of community-based health education, healthcare professionals could disseminate knowledge regarding COVID-19 (e.g., probability of infection and severity of disease) to the general public. The potential benefits of dietary supplements could also be introduced so that the public can decide whether these products would be helpful for the promotion of their immunity against infection. Such measures could help to enhance consumers' positive attitudes and their purchase intention when facing the fear of COVID-19 [90].

## 6. Conclusions

Through incorporating the protection motivation theory (PMT) into the theory of planned behavior (TPB), the current study not only extended the application of TPB but also bridged the knowledge gap between the fear of COVID-19 and the purchase intention toward dietary supplements. Our findings showed that fear of COVID-19 was associated with an enhanced purchase intention toward dietary supplements. Besides, attitudes, subjective norms, and perceived behavioral control were significant factors that mediated the correlation between the fear of COVID-19 and the purchase intention of dietary supplements.

The current study had its limitations. First, since this was a cross-sectional investigation based on a structured questionnaire, a causal relationship between the study parameters could not be established. Further longitudinal studies are warranted to explore the impacts of different parameters (e.g., attitudes, behavioral intention) on the actual behavioral outcomes of consumers. Second, there was an item for assessing the participant's perceived behavioral control, "I have the resource, time, and opportunity to receive vaccination against COVID-19", that encompassed three different criteria, which needed to be concomitantly fulfilled to give a positive response. Therefore, it was impossible to address the precise reason for a negative response that would introduce bias to the analysis. Third, although COVID-19 is a global pandemic affecting up to 222 countries, we could only focus on the city from which the pandemic originated. Although the city has a population of over 8 million, our results were not representative of China or even other cities in China. Further research is warranted to investigate the geographical, ethnic, and cultural impacts on study outcomes. Finally, because the severity of the pandemic and the associated fear vary in different countries, the findings of the present study may not be extrapolated to other countries.

**Supplementary Materials:** The following are available online at https://www.mdpi.com/article/10.3390/su132212900/s1, Table S1: The theory of planned behavior (TPB) questionnaire items.

**Author Contributions:** Conceptualization, C.-C.Y.; writing—original draft, C.L. and C.-C.Y.; writing—review and editing, C.-C.Y., C.-K.S. and S.-Y.Y.; formal analysis, Y.-C.C.; investigation, T.L.; data curation, C.-C.Y. All authors had complete access to the study data that support the publication. All authors have read and agreed to the published version of the manuscript.

**Funding:** This research received no external funding.

**Institutional Review Board Statement:** The study was conducted according to the guidelines of the Declaration of Helsinki, and approved by the institutional review board of Taichung Jen-Ai Hospital (Approval No.: 110-25).

**Informed Consent Statement:** Informed consent was obtained from all subjects involved in the study.

**Data Availability Statement:** The data that support the findings of this study are available on request from the corresponding author. The data are not publicly available due to privacy or ethical restrictions.

**Conflicts of Interest:** The authors declare no conflict of interest.

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
