# Peer review of "The Impact of the Fear of COVID-19 on Purchase Behavior of Dietary Supplements: Integration of the Theory of Planned Behavior and the Protection Motivation Theory"

_sustainability, doi:10.3390/su132212900_

Round 1

Reviewer 2 Report

The Impacts of Fear of COVID-19 on purchase behavior toward 2 dietary supplements. There are several questions and doubts the authors need to address that led me to major revisions. Here are some:

  1. The proposed conceptual framework lacks in-depth discussion. How did the authors come up with the constructs? What makes their model different from similar studies and how did they ensure the validity of the content? There was one comprehensive study by Prasetyo et al., (2020) who also explored factors affecting the perceived effectiveness of COVID-19 prevention measures by utilizing PMT and extended TPB. I’m very surprised that this study failed to discuss the findings of Prasetyo et al., (2020) especially since the model starts with “Fear of COVID-19” since it’s basically the idea of Prasetyo et al., (2020). This manuscript needs to discuss in-depth the differences between Prasetyo et al., (2020) in the introduction and discussion parts.
  • Prasetyo, Y. T.; Castillo, A. M.; Salonga, L. J.; Sia, J. A.; Seneta, J. A. Factors affecting perceived effectiveness of COVID-19 prevention measures among Filipinos during enhanced community quarantine in Luzon, Philippines: Integrating Protection Motivation Theory and extended theory of planned behavior. International Journal of Infectious Diseases 2020, 99, 312–323. https://nam05.safelinks.protection.outlook.com/?url=https%3A%2F%2Fdoi.org%2F10.1016%2Fj.ijid.2020.07.074&data=04%7C01%7Cytprasetyo%40mapua.edu.ph%7C2781c2ff714741f6f3ba08d88bdd4cb0%7Cc7e8b5ac96c64123a65a793543aced4d%7C0%7C0%7C637413129903670551%7CUnknown%7CTWFpbGZsb3d8eyJWIjoiMC4wLjAwMDAiLCJQIjoiV2luMzIiLCJBTiI6Ik1haWwiLCJXVCI6Mn0%3D%7C1000&sdata=L2UPUGnmfP9OhWjeIx39tkUHzweCdcJX3%2BmL7GJmYqo%3D&reserved=0

  1. Similarly, the research gap of this study is lacking in most updated studies. I cannot justify in what way(s) the past researches failed to provide valuable information regarding this matter. Please mention the study below in the research gap and discussion which also utilized structural equation modeling related to purchasing behavior in the COVID-19 context.
  • Prasetyo, Y. T.; Tanto, H.; Mariyanto, M.; Hanjaya, C.; Young, M. N.; Persada, S. F.; Miraja, B. A.; Redi, A. A. Factors affecting customer satisfaction and loyalty in online food delivery service during the COVID-19 pandemic: Its relation with open innovation. Journal of Open Innovation: Technology, Market, and Complexity 2021, 7, 76.
  1. Please show the mean, standard deviation, and factor loading for each indicator.
  2. Please also show each indicator under the latent in one table, not just the description. It will enhance the readability of the paper.
  3. Please show the demographic characteristic of the respondents by using a table, not just the description.
  4. To calculate the validity, it is necessary to calculate the MSV, HTMT and Fornel larcker criterion
  5. The study applies a purposive sampling approach. How did you apply the purposive sampling approach? For example, in some studies, researchers use a "directory" or some kind of database to track/find the respondents. Here, how did you track/find the respondents? In the current version, the sampling approach sounds more like a "convenience sampling approach". You need to clearly spell out how did you find the respondents for applying the purposive sampling approach.
  6. How do you justify the sample size? Is there any particular criterion you followed to select the sample size? For example, if you use a neural network approach, the sample size is not important since the methodology is quite robust to small a sample size. But for the study that uses SEM, how do you justify the sample size?
  7. Which language was used in the questionnaire? Did you use two different versions of the questionnaire? If so, how did ensure both versions are compatible with each other?
  8. Did you use any pilot test before conducting the original survey? Using a pilot test is an important step in SEM to make sure that initial results are satisfactory.
  9. With that super simple model, what is the contribution of your study? The final model is not new if this study is compared to Prasetyo et al., (2020). Again, what’s the biggest contribution of this study if the main idea is published already last year.
  • Prasetyo, Y. T.; Castillo, A. M.; Salonga, L. J.; Sia, J. A.; Seneta, J. A. Factors affecting perceived effectiveness of COVID-19 prevention measures among Filipinos during enhanced community quarantine in Luzon, Philippines: Integrating Protection Motivation Theory and extended theory of planned behavior. International Journal of Infectious Diseases 2020, 99, 312–323. https://nam05.safelinks.protection.outlook.com/?url=https%3A%2F%2Fdoi.org%2F10.1016%2Fj.ijid.2020.07.074&data=04%7C01%7Cytprasetyo%40mapua.edu.ph%7C2781c2ff714741f6f3ba08d88bdd4cb0%7Cc7e8b5ac96c64123a65a793543aced4d%7C0%7C0%7C637413129903670551%7CUnknown%7CTWFpbGZsb3d8eyJWIjoiMC4wLjAwMDAiLCJQIjoiV2luMzIiLCJBTiI6Ik1haWwiLCJXVCI6Mn0%3D%7C1000&sdata=L2UPUGnmfP9OhWjeIx39tkUHzweCdcJX3%2BmL7GJmYqo%3D&reserved=0

Reviewer 3 Report

Dear authors,

The article is clear and relevant to the field.

The study aimed to assess the impacts of fear of COVID-19 on consumer purchasing behavior about dietary supplements. I find the study of dietary supplements in the context of the Pandemic very interesting because I do not identify similar studies. The main contributions of the article were to demonstrate that attitudes, subjective norms and perceived behavioral control had a positive impact on the intention to buy dietary supplements.

The sample also proved that fear of COVID-19 was related to a purchase intention about dietary supplements.

Attitudes, subjective norms and perceived behavioral control were significant factors that mediated the association between fear of COVID-19 and intention to purchase dietary supplements.

This study makes interesting contributions to the definition of marketing/communication strategies for dietary supplements in times of Pandemic.

It was interesting to combine the Protection Motivation Theory with the Planned Behavior Theory to study the impact of the COVID-19 Pandemic on the intention to buy dietary supplements. It seems that cultural variables should have been studied, given the tremendous cultural differences between different countries.

Bibliographical references are comprehensive and adequate.

It is necessary to review the numbering of citations. No. 40 does not exist. The text reads: "Attitude is an individual's positive or negative evaluation of a specific behavior. The more positive the attitudes, the higher the behavior intention, and vice versa. A number of previous studies have shown a positive association between attitudes and purchase intention [39-41]."

The conclusions are consistent with the results obtained in the empirical study and the literature review.

Ethics statements and data availability statements are adequate.

Round 2

Reviewer 1 Report

The authors have followed my suggestions and now, I have no concerns. I recommend the publication of this article. 

Reviewer 2 Report

This paper is substantially improved.